# Gender differences in the Italian academic landscape: Examining inequalities within the medical area in the last decade

Roberta Magnano San Lio[1,2], Mara Morini[2], Enrico Di Rosa[2,3], Alessandra Sinopoli [2,3], Virginia Casigliani[2,4], Manuela Martella [2,5], Andrea Maugeri[1,2], Martina Barchitta[1,2], Maria Carmela Agodi[6], Ilenia Picardi[6], Antonella Agodi [1,2]*

1 Department of Medical and Surgical Sciences and Advanced Technologies "GF Ingrassia", University of Catania, Catania, Italy, 2 Gender Prevention Working Group, Italian Society of Hygiene, Preventive Medicine, and Public Health (SItI), Rome, Italy, 3 Department of Prevention, Local Health Authority Roma 1, Rome, Italy, 4 Department of Translational Research and of New Surgical and Medical Technologies, University of Pisa, Pisa, Italy, 5 Department of Public Health Science and Pediatrics, University of Turin, Torino, Italy, 6 Department of Political Science, University of Naples Federico II, Naples, Italy

* antonella.agodi@unict.it

## Abstract

Gender equality remains a key focus in research and innovation policies. However, the academic sector still exhibits significant under-representation of women in leadership roles and horizontal segregation affecting career trajectories. This study examines gender inequalities in top positions and career progression within Italy's Medical Sciences sector (Area 06) from 2014 to 2023, through data from the Cineca portal, including details on academic roles, gender, region, and scientific-disciplinary sectors (SSDs). Gender disparity was evaluated using the Glass Ceiling Index (GCI) and the Glass Door Index (GDI). Data analysis up to 2023 shows a gender inversion in career progression: women dominate early career stages – accounted for 71.5% of research fellows and 57.1% of RTDA positions –, but men prevail in senior roles, with 61.8% associate professors and 71.8% full professors. This trend is consistent across most SSDs. Although temporal analysis indicates a reduction in gender disparities in top positions – with a GCI ranging from 2.3 in 2015 to 1.7 in 2023 – GCI values above one across all regions suggest persistent gender inequality. The GDI values, ranging from 1.3 in 2015 to 1.6 in 2023, exhibit considerable variability; however, overall, the disparities have worsened by 2023. These findings underscore the need for concrete measures to enhance equity and inclusivity in academia. Developing and implementing initiatives, programs, and policies that guarantee equal opportunities and resources for all researchers, irrespective of gender, is essential.

**Data availability statement:** All relevant data are within the manuscript and its Supporting Information files.

**Funding:** This work was partially funded by the University of Catania, Italy, Department of Medical and Surgical Science and Advanced Technologies 'GF Ingrassia' (UPB: 5C130011075), awarded to A.M.

**Competing interests:** The authors have declared that no competing interests exist.

**Abbreviations:** GCI, Glass Ceiling Index; GDI, Glass Door Index; RTDA, Fixed-term Researcher Type A; RTDB, Fixed-term Researcher Type B; SSD, Scientific-disciplinary sector (Settore scientifico-disciplinare).

## 1. Introduction

Gender inequality remains a persistent issue in various professional fields, and academia is no exception [1,2]. Despite regulatory interventions and gender mainstreaming policies, gender segregation persists, leading to the continued underrepresentation of women in top positions [3]. The European context provides a broader framework for analysing these issues, as gender disparity in academia is a well-documented phenomenon across the continent. According to the European Commission's "She Figures 2021" report, women remain underrepresented in scientific and technical fields, and their progression to senior academic roles is significantly hindered compared to their male counterparts [4]. Disparities are particularly evident in STEM (science, technology, engineering, and mathematics) fields, where women remain a minority in both education and employment. Financial disparities also persist, as women researchers are less likely to receive competitive grants or lead major projects. However, the composition of teams – particularly in terms of gender balance – can significantly enhance collective performance. Evidence suggests that gender diversity strengthens the effectiveness of workgroups within STEM fields. Therefore, it is essential not only to promote greater access for women to education and careers in STEM, but also to encourage a transformation in social and professional dynamics. This shift should aim to recognize gender diversity as a key driver of innovation, creativity, and sustainable growth across these sectors [5].

The Gender Equality Strategy 2020–2025 – adopted by the European Commission in March 2020 to set out the priorities and actions the EU intends to undertake during the 2020–2025 period to reduce gender inequalities – aims to create an ecosystem where women and men have equal opportunities to thrive in all aspects of academic and scientific life, contributing to a more inclusive and innovative Europe [6]. The gender gaps – especially in academia and research – reflect structural and cultural barriers, including unconscious biases, work-life balance challenges, and limited access to mentorship and networking opportunities. Addressing these disparities requires comprehensive approaches, such as targeted funding for women-led research, policies supporting work-life balance, and initiatives fostering inclusivity in traditionally male-dominated disciplines. In practice, these measures have already been incorporated into the Horizon Europe programme, which not only dedicates specific calls for proposals aimed at promoting gender equality in research teams, but also requires participating institutions to have a Gender Equality Plan (GEP) in place as an eligibility criterion for funding. Furthermore, efforts are being made to enhance the visibility of female role models in academia, which can provide inspiration and guidance to younger generations of women scientists [7].

In Italy, these disparities are evident in numerous aspects of academic life, including hiring practices, career progression, and access to research funding [8]. Recent research and reports indicate that, while women constitute a significant proportion of university graduates in Italy, their presence still dwindles at higher academic ranks [9]. These inequalities also include gender-based differentiation throughout academic career paths in the current Italian university system, beginning from the

recruitment stages [10–13]. Recent studies have revealed that, despite women often having more distinguished graduate and post-graduate careers and earning degrees faster than men, especially in STEM fields, they are predominantly found in temporary positions in academia [14]. The percentage of female researchers hired on a permanent basis is significantly lower compared to their male counterparts, as it also happens at the European level.

In Italy, academic careers in research and teaching are structured into several roles: Full Professor, Associate Professor, Researcher (a position being phased out), Fixed-term Researcher Type B (RTDB) as per Art. 24, para. 3, letter b of Law 240/2010, Fixed-term Researcher Type A (RTDA) as per Art. 24, para. 3, letter a of Law 240/2010, and Research Fellow.

RTDB positions involve a non-renewable three-year contract, but researchers can transition directly to Associate Professor if they possess the National Scientific Qualification and receive a positive evaluation from the university (it is a tenure track position). RTDA positions have a contract lasting three years, renewable for an additional two years. Research fellows have contracts ranging from a minimum of one year to a maximum of three years, with a cumulative limit of six years for an individual in this role. As per the MIUR decree *"Integrazione del decreto del Ministro dell'istruzione, dell'università e della ricerca 1 settembre 2016, n. 662, recante la tabella di corrispondenza tra posizioni accademiche italiane ed estere ai sensi dell'articolo 18, comma 1, lettera b), della Legge 30 dicembre 2010, n. 240."*, which specifies the correspondence between Italian and foreign academic positions, full professors correspond to GRADE A, associate professors to GRADE B, RTDB to GRADE C, and RTDA to GRADE D [15], as these GRADES are defined in the SHE Figures Report classification [4].

In addition to the roles mentioned, the position of Temporary Extraordinary Professor (as per Art. 1, para. 12 of Law 230/2005) remains in effect. This role involves a three-year contract that can be renewed for an additional three years and is funded by external entities for research activities. It is reserved for those who have obtained eligibility for the full professorship position or for individuals with high scientific and professional qualifications. Holders of these positions are entitled, for the duration of the contract, to the legal and economic treatment of full professors, with possible additional economic supplements, if provided for by the agreement [16].

The academic landscape in Italy has faced growing scrutiny over gender disparities in recent years, particularly within the medical field (Area 06 – Medicine). This study seeks to comprehensively analyse gender inequalities in academic medicine over the past decade, focusing on the period following the Gelmini Reform (Law 240/2010). This reform fundamentally reshaped the Italian academic system by introducing two fixed-term positions—the fixed-term contract A (RTDA) and the tenure-track fixed-term contract B (RTDB)—while abolishing the previously permanent role of researcher.

The primary aim of this study is to examine how these structural changes have influenced gender representation across different academic roles within medicine. It also aims to identify disparities across scientific-disciplinary sectors (Settori Scientifico-Disciplinari, SSD) and geographic regions in Italy, shedding light on regional and sector-specific dynamics that impact gender equity in academic career progression. By doing so, the study provides a nuanced understanding of the systemic challenges and opportunities for promoting gender parity in Italian academic medicine.

## 2. Materials and methods

The study analysed comprehensive data on academic roles spanning the years 2014–2023, sourced from the Cineca portal (https://cercauniversita.mur.gov.it/php5/docenti/cerca.php?SESSION). This dataset included detailed records on academic positions, encompassing research fellows, fixed-term researchers (RTDA, RTDB), tenured researchers, associate professors, full professors, and temporary extraordinary professors. Each record provided information on academic roles, gender, affiliated university, region, and scientific-disciplinary sectors (SSD). Notably, data for research fellows were limited to the year 2023, while data for other positions spanned the entire period under study.

Certain categories of researchers were excluded from this analysis to ensure consistency and clarity. Specifically, data on full-time fixed-term researchers introduced under Law 79/2022, which replaced research fellowships in 2022, as well as researchers employed under Art. 1, para. 14, Law 230/05 prior to the differentiation into RTDA and RTDB categories,

and data on non-tenured researchers, were omitted. This exclusion was necessary to maintain a uniform approach to analysing academic roles and to focus on positions with clear and consistent categorizations over the period.

For each academic role, data were initially examined by calculating gender proportions and the female-to-male ratio, providing insights into the distribution of men and women across various positions. To assess gender disparities, particularly in senior academic positions and recruitment pathways, the Glass Ceiling Index (GCI) was utilized. The GCI serves as a relative measure, comparing the proportion of women in academia at all levels (grades A, B, C, and D) to their representation in top academic positions (grade A) for a given year.

The GCI is computed using key gender-disaggregated variables for a given year Y, as follows:

- $F_{AY}$: Number of women full professors.

- $M_{AY}$: Number of men full professors.

- $F_{BY}$: Number of women associate professors.

- $M_{BY}$: Number of men associate professors.

- $F_{CY}$ and $F_{DY}$: Number of women researchers (RTDB and RTDA, respectively).

- $M_{CY}$ and $M_{DY}$: Number of men researchers (RTDB and RTDA, respectively).

These variables allow for a detailed examination of gender distribution across the academic hierarchy, from researchers to senior professors. The GCI quantitatively evaluates the extent to which women are underrepresented in higher academic positions relative to their overall presence in academia.

According to the She Figures Report, the GCI is defined as follows [17]:

$$GCI = (F_{AY} + F_{BY} + F_{CY} + F_{DY}) / (F_{AY} + F_{BY} + F_{CY} + F_{DY} + M_{AY} + M_{BY} + M_{CY} + M_{DY}) / F_{AY} / (F_{AY} + M_{AY})$$

Where the numerator calculates the proportion of women among the total academic workforce (all grades combined: A, B, C, and D), accounting for both female ($F_{AY}$, $F_{BY}$, $F_{CY,}$ and $F_{DY}$) and male ($M_{AY}$, $M_{BY}$, $M_{CY,}$ and $M_{DY}$) staff. The denominator calculates the proportion of women at the grade A level ($F_{AY}$) relative to the total number of grade A professors ($F_{AY}$ and $M_{AY}$). Accordingly, the GCI is a continuous metric ranging from 0 to infinity, providing insights into gender disparities in achieving the highest academic ranks. A GCI of 1 indicates gender parity in promotion opportunities. It indicates that women and men have equal chances of being promoted to grade A (Full Professor), reflecting a balanced academic system. A GCI below 1 suggests that women are overrepresented at grade A relative to their overall presence in academia. This scenario is rare and might occur in specific disciplines or contexts where targeted policies or historical trends have favoured women's advancement. A GCI above 1 reveals a glass ceiling effect, where women are underrepresented in grade A positions compared to their overall representation in academia. The higher the GCI value, the more significant the barrier for women in advancing to Full Professor roles.

The more innovative part of the analysis is the one quantitatively analysing gender disparities in the recruitment phase, where the effects of the introduction of the fixed term positions RTDA and RTDB deploys its effects. To this aim, we employed the Glass Door Index (GDI), an index recently introduced to reveal and measure the gender bias in academic recruitment [18]. It is calculated as the ratio between two percentages: the percentage of women in temporary research positions ($PW_{\leq D}$) and the percentage of women in positions that lead to stable academic careers ($PW_D$). This index provides insight into the progression of women from temporary roles to more permanent, stable academic positions. Specifically:

$$GDI = PW_{\leq D,Y} / PW_{DY} = F_{\leq D,Y} / F_{\leq D,Y} + M_{\leq D,Y} / F_{DY} / F_{DY} + M_{DY}$$

where $F_{DY}$ ($M_{DY}$) denotes the number of women (men) in the position leading to stable academic roles in year Y, and $F_{\leq D,Y}$ ($M_{\leq D,Y}$) represents the number of women (men) in temporary research positions and the position leading to academic roles in year Y. Specifically, we computed the GDI value using the RTDB position as the initial stable academic role. Briefly, based on the relative presence of women in temporary research roles and positions leading to academic roles, the GDI measures the fraction that has attained stabilization. Like the GCI, the GDI ranges from 0 to infinity. A GDI equal to or less than 1 indicates that the percentage of women in the tenure track position (RTDB) is greater or equal compared to their percentage in temporary positions (RTDA). Conversely, a GDI above 1 signals the presence of a glass door that restricts women's entry, reflecting a recruitment process towards permanent positions in academia biased against women. The higher the GDI value, the stronger the impact of this glass door on entry into academia. It is important to note that in certain cases, the calculation of the GCI and GDI was not feasible due to the absence of key academic roles in the available data. Specifically, when one or more relevant positions were entirely missing, the indices could not be computed reliably. For instance, the absence of full professors made it impossible to calculate the GCI, which relies on comparing the representation of women at the highest academic rank to their overall presence. Similarly, the GDI could not be determined in cases where either RTDA or RTDB positions were not present, as both are essential to assess transitions from temporary to permanent academic roles.

In 2023, our analysis employed a variety of methodological approaches to comprehensively investigate gender representation across academic levels. Descriptive statistics were used to calculate the percentages of males and females at each academic rank, alongside the computation of the GDI and the Glass Ceiling Index GCI to quantitatively measure disparities in recruitment and progression. To visualize these disparities, we utilized scissor diagrams to depict gender distributions across academic levels, illustrating the divergence in career trajectories between men and women. Geographic variations in gender disparities were analysed through choropleth maps, which displayed the GDI and GCI indices across Italy, providing a spatial representation of regional differences (i.e., the region Valle d'Aosta is not present because it does not host any university). Maps were generated using R (version 4.3.2) and the packages rnaturalearth (v0.3.3), rnaturalearthdata (v0.1.0), sf (v1.0-14), and ggplot2 (v3.4.4). Geographic data are from Natural Earth (https://www.naturalearthdata.com) and are in the public domain.

The temporal dimension of gender disparities was explored using a longitudinal analysis covering the period from 2014 to 2023. This analysis employed line graphs to examine changes in GDI, GCI, and the female-to-male ratio over time, allowing for the identification of trends and shifts across the decade. Heatmaps were further applied to capture the evolution of GDI and GCI across scientific-disciplinary sectors (*Settori Scientifico Disciplinari*, SSD) and Italian regions, enabling a granular understanding of gender disparities within specific fields and locations. The SSD classification used in this analysis was aligned with the official categorizations provided by the Italian National University Council (CUN; https://www.cun.it/uploads/storico/settori_scientifico_disciplinari_english.pdf), as detailed in Table S1 in S1 File. Together, these methodological tools provided a robust framework for analysing and visualizing the complex dynamics of gender representation and disparities in academia.

## 3. Results

### 3.1. Gender disparities in Italian academic Medical Area in 2023

In 2023, women accounted for a total of 4,783 individuals across all academic roles, while men numbered 6,353, highlighting a marked gender imbalance. This imbalance is particularly evident when analyzing the distribution across different career stages. Women were more represented in early-career positions, such as research fellows (1,157 women vs. 462 men) and RTDA positions (751 women vs. 565 men). However, as academic rank increases, the trend reverses: men outnumber women in RTDB positions (625 men vs. 431 women), among researchers (557 men vs. 491 women), associate professors (2,371 men vs. 1,465 women), and even more so among full professors (1,722 men vs. 483 women). The disparity reaches its peak among temporary extraordinary professors, with 51 men compared to just 5 women.

This pattern is further illustrated by the "career scissor" diagram, a widely used representation of vertical segregation in academia. The diagram clearly shows the predominance of women in the initial stages of the academic path – 71.5% of research fellows and 57.1% of RTDA positions are held by women. However, the proportion of women steadily declines at each successive stage: 40.8% in RTDB roles, 46.8% among researchers, 38.2% among associate professors, and just 21.9% at the full professor level, compared to 78.1% men (Fig 1).

The greatest disparity is observed among temporary extraordinary professors, with 91.1% men and only 8.9% women (Fig 2).

Gender disparity is prevalent across most SSDs, with absolute numbers reported in Table S2 in S1 File. Specifically, a gender segregation with a clear predominance of men was observed for the following SSDs: MED/10 – Respiratory

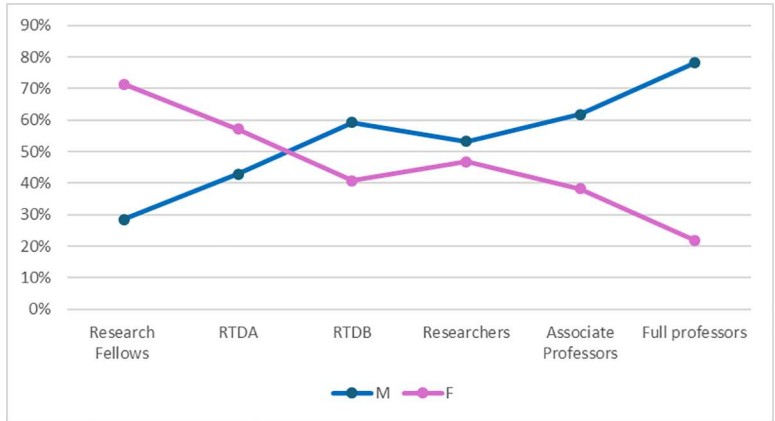

**Fig 1.  Career scissor diagram in Italian Medical Area in 2023.**

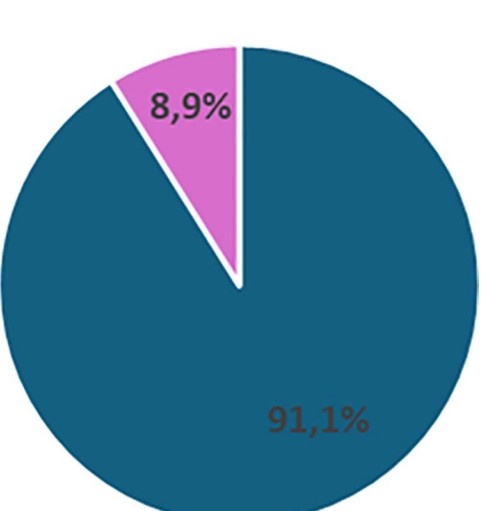

**Fig 2.  Gender distribution of Temporary Extraordinary Professor in Italian Medical Area in 2023.**

diseases, MED/11 – Cardiovascular diseases, MED/13 – Endocrinology, MED/14 – Nephrology, MED/18 – General surgery, MED/19 – Plastic surgery, MED/20 – Paediatric surgery, MED/21 – Thoracic surgery, MED/22 – Vascular surgery, MED/23 – Heart surgery, MED/24 – Urology, MED/27 – Neurosurgery, MED/28 – Oral diseases and dentistry, MED/29 – Maxillofacial surgery, MED/30 – Eye diseases, MED/31 – Otorhinolaryngology, MED/33 – Musculoskeletal system diseases, MED/37 – Neuroradiology, MED/40 – Obstetrics and gynaecology, MED/41 – Anaesthesiology. Conversely, there were other fields where gender segregation is less pronounced, and women equal or surpass the percentage of men (e.g., MED/01 – Medical statistics, MED/02 – Medical history, MED/03 – Medical genetics, MED/04 – Experimental medicine and pathophysiology, MED/05 – Clinical pathology, MED/07 – Microbiology and clinical microbiology, MED/38 – General and subspecialty paediatrics, MED/39 – Child neuropsychiatry, MED/42 – Hygiene and public health, MED/45 – Nursing sciences: general, clinical and paediatric, MED/47 – Medical and biotechnology laboratory techniques, MED/48 – Midwifery, MED/49 – Neuropsychiatric and rehabilitation nursing sciences, MED/50 – Food sciences and dietetics). However, for the SSDs MED/38, MED/39, MED/42, MED/45, MED/49, and MED/50, there continues to be a predominance of men in the role of full professor. Moreover, in general, women were more frequently represented in Research Fellow and RTDA roles but were underrepresented at higher academic levels. However, this pattern does not hold in SSDs MED/22, MED/24, MED/27, MED/33, and MED/37, where men outnumber women in all academic positions. Notably, in SSD MED/20, men held a larger proportion of Research Fellow positions, while women were predominantly found in RTDA roles. Further details are shown in Table 1.

### 3.2. Gender disparities across Italian regions in 2023

We also analysed gender disparity across Italian regions using the GCI and GDI, while absolute figures for each academic position are provided in Table S3 in S1 File. Our findings indicate a presence of a glass ceiling effect in all Italian regions, suggesting that women are underrepresented in Full Professor positions compared to other academic roles. Excluding regions for which the available data did not fulfil the conditions necessary for the calculation of the GCI (i.e., Basilicata, Molise, and Valle d'Aosta), Sardinia, Liguria, and Veneto displayed the highest levels of gender inequality, disadvantaging women, with GCI values of 2.9, 2.5, and 2.3, respectively. Conversely, the lowest GCI values were observed in Piedmont (1.4), Trentino-Alto Adige (1.3), and Friuli-Venezia Giulia (1.2) (Fig 3).

According to the GDI, only the regions of Molise and Trentino-Alto Adige reported a value below one, highlighting that in the tenure track position, the percentage of women is higher (or equal) compared to the percentage of women in temporary positions. This result must be considered cautiously as it could be only due to the low number of professionals in the academic roles in these regions. By contrast, the highest GDI values were detected in Abruzzo (2.7), Apulia (2.2), Sardinia (2.1) (Fig 4).

### 3.3. Gender disparities in Italian academic Medical Area from 2014 to 2023

We next assessed the temporal trend of female-to-male ratio from 2015 to 2023, according to academic roles (Fig 5). Over time, there has been a gradual rise in female-to-male ratios, suggesting an increasing presence of women across all academic positions. This increase has been moderate across most roles, apart from RTDA positions, where a more significant rise has been observed. It is worthwhile to note that gender parity (a female-to-male ratio of 1) has not been achieved, except in the RTDA category.

We next assessed the temporal trend – from 2015 to 2023 – of gender disparities in Italian academic Medical Area. According to GCI, the temporal analysis showed a reduction in female disparities in academic top positions, from 2.3 in 2015 to 1.7 in 2023. Moreover, GDI increased from 1.3 in 2015 to 1.6 in 2023 (Fig 6).

We also examined temporal trends in gender disparities within regions using GDI and GCI. In general, the GCI remained above 1 across all regions and throughout the entire period under analysis, with the exception of Molise and Trentino-Alto Adige in 2017, which recorded values of 0.3 and 0.8, respectively. However, it is important to highlight that,

**Table 1. Gender disparity within SSDs.**

| SSD | | Gender | Research Fellows | RTDA | RTDB | Tenured Researches | Associate Professors | Full Professors | Temporary Extraordinary Professor |
|---|---|---|---|---|---|---|---|---|---|
| MED/01 | Medical statistics | Female | 73.3% | 68.8% | 50.0% | 50.0% | 63.8% | 51.2% | |
| | | Male | 26.7% | 31.3% | 50.0% | 50.0% | 36.2% | 48.8% | |
| MED/02 | Medical history | Female | 80.0% | 66.7% | 50.0% | 40.0% | 46.7% | 66.7% | |
| | | Male | 20.0% | 33.3% | 50.0% | 60.0% | 53.3% | 33.3% | |
| MED/03 | Medical genetics | Female | 83.9% | 64.7% | 68.4% | 82.4% | 58.2% | 39.6% | |
| | | Male | 16.1% | 35.3% | 31.6% | 17.6% | 41.8% | 60.4% | |
| MED/04 | Experimental medicine and pathophysiology | Female | 76.0% | 74.7% | 63.9% | 72.9% | 61.9% | 40.8% | 0.0% |
| | | Male | 24.0% | 25.3% | 36.1% | 27.1% | 38.1% | 59.2% | 100.0% |
| MED/05 | Clinical pathology | Female | 85.0% | 88.9% | 72.7% | 73.3% | 70.5% | 46.7% | |
| | | Male | 15.0% | 11.1% | 27.3% | 26.7% | 29.5% | 53.3% | |
| MED/06 | Medical oncology | Female | 76.7% | 50.0% | 40.0% | 50.0% | 30.6% | 15.0% | 33.3% |
| | | Male | 23.3% | 50.0% | 60.0% | 50.0% | 69.4% | 85.0% | 66.7% |
| MED/07 | Microbiology and clinical microbiology | Female | 69.4% | 76.3% | 55.6% | 63.6% | 71.6% | 50.0% | |
| | | Male | 30.6% | 23.7% | 44.4% | 36.4% | 28.4% | 50.0% | |
| MED/08 | Pathology | Female | 63.3% | 50.0% | 52.4% | 57.7% | 44.9% | 32.0% | |
| | | Male | 36.7% | 50.0% | 47.6% | 42.3% | 55.1% | 68.0% | |
| MED/09 | Internal medicine | Female | 68.4% | 48.3% | 36.2% | 51.4% | 33.9% | 16.9% | 0.0% |
| | | Male | 31.6% | 51.7% | 63.8% | 48.6% | 66.1% | 83.1% | 100.0% |
| MED/10 | Respiratory diseases | Female | 81.0% | 64.3% | 30.0% | 37.5% | 39.5% | 19.4% | 0.0% |
| | | Male | 19.0% | 35.7% | 70.0% | 62.5% | 60.5% | 80.6% | 100.0% |
| MED/11 | Cardiovascular diseases | Female | 76.3% | 54.8% | 19.4% | 35.5% | 29.7% | 8.2% | 0.0% |
| | | Male | 23.7% | 45.2% | 80.6% | 64.5% | 70.3% | 91.8% | 100.0% |
| MED/12 | Gastroenterology | Female | 76.7% | 45.5% | 50.0% | 71.4% | 26.6% | 19.0% | 0.0% |
| | | Male | 23.3% | 54.5% | 50.0% | 28.6% | 73.4% | 81.0% | 100.0% |
| MED/13 | Endocrinology | Female | 87.1% | 60.0% | 63.6% | 35.7% | 48.5% | 19.0% | |
| | | Male | 12.9% | 40.0% | 36.4% | 64.3% | 51.5% | 81.0% | |
| MED/14 | Nephrology | Female | 83.3% | 47.1% | 50.0% | 50.0% | 27.3% | 14.3% | 50.0% |
| | | Male | 16.7% | 52.9% | 50.0% | 50.0% | 72.7% | 85.7% | 50.0% |
| MED/15 | Blood diseases | Female | 67.6% | 69.0% | 36.4% | 52.6% | 41.0% | 12.5% | 0.0% |
| | | Male | 32.4% | 31.0% | 63.6% | 47.4% | 59.0% | 87.5% | 100.0% |
| MED/16 | Rheumatology | Female | 80.0% | 61.5% | 30.0% | 66.7% | 54.4% | 19.2% | |
| | | Male | 20.0% | 38.5% | 70.0% | 33.3% | 45.6% | 80.8% | |
| MED/17 | Infectious diseases | Female | 77.8% | 50.0% | 38.9% | 56.0% | 37.9% | 25.0% | 0.0% |
| | | Male | 22.2% | 50.0% | 61.1% | 44.0% | 62.1% | 75.0% | 100.0% |
| MED/18 | General surgery | Female | 76.2% | 30.4% | 23.9% | 23.1% | 11.3% | 3.5% | 0.0% |
| | | Male | 23.8% | 69.6% | 76.1% | 76.9% | 88.7% | 96.5% | 100.0% |
| MED/19 | Plastic surgery | Female | 66.7% | 40.0% | 26.7% | 20.0% | 12.5% | 3.8% | 0.0% |
| | | Male | 33.3% | 60.0% | 73.3% | 80.0% | 87.5% | 96.2% | 100.0% |
| MED/20 | Paediatric surgery | Female | 33.3% | 66.7% | 33.3% | 40.0% | 9.5% | 14.3% | |
| | | Male | 66.7% | 33.3% | 66.7% | 60.0% | 90.5% | 85.7% | |
| MED/21 | Thoracic surgery | Female | 100.0% | 50.0% | 50.0% | 0% | 7.7% | 5.3% | |
| | | Male | 0% | 50.0% | 50.0% | 100.0% | 92.3% | 94.7% | |
| MED/22 | Vascular surgery | Female | 0% | 44.4% | 0% | 18.2% | 12.2% | 0% | |
| | | Male | 100.0% | 55.6% | 100.0% | 81.8% | 87.8% | 100.0% | |

*(Continued)*

| SSD | | Gender | Research Fellows | RTDA | RTDB | Tenured Researches | Associate Professors | Full Professors | Temporary Extraordinary Professor |
|---|---|---|---|---|---|---|---|---|---|
| MED/23 | Heart surgery | Female | 66.7% | 57.1% | 0% | 0% | 2.4% | 3.6% | |
| | | Male | 33.3% | 42.9% | 100.0% | 100.0% | 97.6% | 96.4% | |
| MED/24 | Urology | Female | 40.0% | 7.1% | 6.7% | 0% | 2.1% | 2.2% | |
| | | Male | 60.0% | 92.9% | 93.3% | 100.0% | 97.9% | 97.8% | |
| MED/25 | Psychiatry | Female | 60.6% | 40.9% | 38.5% | 37.5% | 43.1% | 24.4% | 0.0% |
| | | Male | 39.4% | 59.1% | 61.5% | 62.5% | 56.9% | 75.6% | 100.0% |
| MED/26 | Neurology | Female | 64.0% | 47.0% | 39.5% | 50.0% | 41.1% | 17.4% | 100.0% |
| | | Male | 36.0% | 53.0% | 60.5% | 50.0% | 58.9% | 82.6% | 0.0% |
| MED/27 | Neurosurgery | Female | 37.5% | 20.0% | 12.5% | 16.7% | 5.1% | 3.1% | 0.0% |
| | | Male | 62.5% | 80.0% | 87.5% | 83.3% | 94.9% | 96.9% | 100.0% |
| MED/28 | Oral diseases and dentistry | Female | 60.4% | 31.3% | 31.7% | 36.8% | 22.2% | 21.5% | 0.0% |
| | | Male | 39.6% | 68.8% | 68.3% | 63.2% | 77.8% | 78.5% | 100.0% |
| MED/29 | Maxillofacial surgery | Female | 50.0% | 14.3% | 22.2% | 18.2% | 14.3% | 0% | 0.0% |
| | | Male | 50.0% | 85.7% | 77.8% | 81.8% | 85.7% | 100.0% | 100.0% |
| MED/30 | Eye diseases | Female | 56.0% | 41.4% | 53.8% | 39.0% | 24.7% | 4.7% | 0.0% |
| | | Male | 44.0% | 58.6% | 46.2% | 61.0% | 75.3% | 95.3% | 100.0% |
| MED/31 | Otorhinolaryngology | Female | 90.9% | 22.2% | 17.6% | 23.5% | 13.6% | 0% | 0.0% |
| | | Male | 9.1% | 77.8% | 82.4% | 76.5% | 86.4% | 100.0% | 100.0% |
| MED/32 | Audiology | Female | 75.0% | 16.7% | 100.0% | 62.5% | 36.7% | 37.5% | |
| | | Male | 25.0% | 83.3% | 0% | 37.5% | 63.3% | 62.5% | |
| MED/33 | Musculoskeletal system diseases | Female | 33.3% | 22.2% | 3.2% | 11.1% | 9.3% | 0% | 0.0% |
| | | Male | 66.7% | 77.8% | 96.8% | 88.9% | 90.7% | 100.0% | 100.0% |
| MED/34 | Physical and rehabilitation medicine | Female | 43.8% | 54.5% | 40.0% | 57.1% | 35.1% | 25.9% | |
| | | Male | 56.3% | 45.5% | 60.0% | 42.9% | 64.9% | 74.1% | |
| MED/35 | Dermatological and venereological diseases | Female | 65.6% | 62.5% | 68.2% | 64.3% | 46.6% | 32.3% | 0.0% |
| | | Male | 34.4% | 37.5% | 31.8% | 35.7% | 53.4% | 67.7% | 100.0% |
| MED/36 | Diagnostic imaging and radiotherapy | Female | 62.2% | 35.9% | 24.4% | 50.0% | 35.7% | 16.3% | 100.0% |
| | | Male | 37.8% | 64.1% | 75.6% | 50.0% | 64.3% | 83.7% | 0.0% |
| MED/37 | Neuroradiology | Female | 0% | 20.0% | 10.0% | 0% | 31.6% | 21.1% | 0.0% |
| | | Male | 100.0% | 80.0% | 90.0% | 100.0% | 68.4% | 78.9% | 100.0% |
| SSD | | Gender | Research Fellows | RTDA | RTDB | Tenured Researches | Associate Professors | Full Professors | Temporary Extraordinary Professor |
| MED/38 | General and subspecialty paediatrics | Female | 76.4% | 73.8% | 51.4% | 58.8% | 51.6% | 23.3% | 50.0% |
| | | Male | 23.6% | 26.2% | 48.6% | 41.2% | 48.4% | 76.7% | 50.0% |
| MED/39 | Child neuropsychiatry | Female | 78.9% | 91.7% | 68.8% | 72.7% | 64.9% | 30.0% | |
| | | Male | 21.1% | 8.3% | 31.3% | 27.3% | 35.1% | 70.0% | |
| MED/40 | Obstetrics and gynaecology | Female | 88.9% | 46.2% | 44.8% | 63.4% | 38.1% | 13.1% | 0.0% |
| | | Male | 11.1% | 53.8% | 55.2% | 36.6% | 61.9% | 86.9% | 100.0% |
| MED/41 | Anaesthesiology | Female | 53.3% | 41.7% | 30.0% | 45.9% | 23.9% | 12.5% | 0.0% |
| | | Male | 46.7% | 58.3% | 70.0% | 54.1% | 76.1% | 87.5% | 100.0% |
| MED/42 | Hygiene and public health | Female | 75.0% | 63.0% | 53.6% | 65.0% | 63.2% | 36.8% | 0.0% |
| | | Male | 25.0% | 37.0% | 46.4% | 35.0% | 36.8% | 63.2% | 100.0% |
| MED/43 | Forensic medicine | Female | 76.5% | 35.3% | 36.8% | 40.0% | 37.8% | 23.3% | 0.0% |
| | | Male | 23.5% | 64.7% | 63.2% | 60.0% | 62.2% | 76.7% | 100.0% |

*(Continued)*

**Table 1.** (Continued)

| SSD | | Gender | Research Fellows | RTDA | RTDB | Tenured Researches | Associate Professors | Full Professors | Temporary Extraordinary Professor |
|---|---|---|---|---|---|---|---|---|---|
| **MED/44** | Occupational medicine | Female | 52.0% | 53.3% | 35.7% | 60.0% | 44.4% | 25.0% | 0.0% |
| | | Male | 48.0% | 46.7% | 64.3% | 40.0% | 55.6% | 75.0% | 100.0% |
| **MED/45** | Nursing sciences: general, clinical and paediatric | Female | 88.9% | 57.9% | 28.6% | | 56.7% | 75.0% | |
| | | Male | 11.1% | 42.1% | 71.4% | | 43.3% | 25.0% | |
| **MED/46** | Medical and biotechnol-ogy laboratory techniques | Female | 76.4% | 87.7% | 64.3% | 70.0% | 70.1% | 52.8% | |
| | | Male | 23.6% | 12.3% | 35.7% | 30.0% | 29.9% | 47.2% | |
| **MED/47** | Midwifery | Female | | 100.0% | | 100.0% | 100.0% | | |
| | | Male | | 0.0% | | 0.0% | 0.0% | | |
| **MED/48** | Neuropsychiatric and rehabilitation nursing sciences | Female | 55.6% | 70.0% | 10.0% | 33.3% | 73.3% | 57.1% | |
| | | Male | 44.4% | 30.0% | 90.0% | 66.7% | 26.7% | 42.9% | |
| **MED/49** | Food sciences and dietetics | Female | 77.1% | 73.7% | 77.3% | 45.5% | 52.5% | 34.6% | |
| | | Male | 22.9% | 26.3% | 22.7% | 54.5% | 47.5% | 65.4% | |
| **MED/50** | Applied medical techniques | Female | 64.1% | 79.2% | 53.6% | 45.5% | 45.5% | 28.9% | 0.0% |
| | | Male | 35.9% | 20.8% | 46.4% | 54.5% | 54.5% | 71.1% | 100.0% |

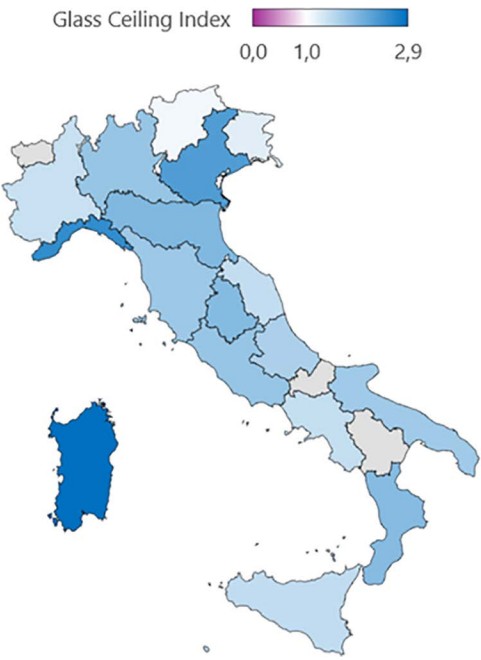

**Fig 3. Gender disparity in Italian Medical Area in 2023 according to Glass Ceiling Index. Map created using R (version 4.3.2) and the packages rnaturalearth (v0.3.3), rnaturalearthdata (v0.1.0), sf (v1.0-14), and ggplot2 (v3.4.4). Geographic data are from Natural Earth (**https://www.naturale-arthdata.com**) and are in the public domain. No copyright restrictions apply.**

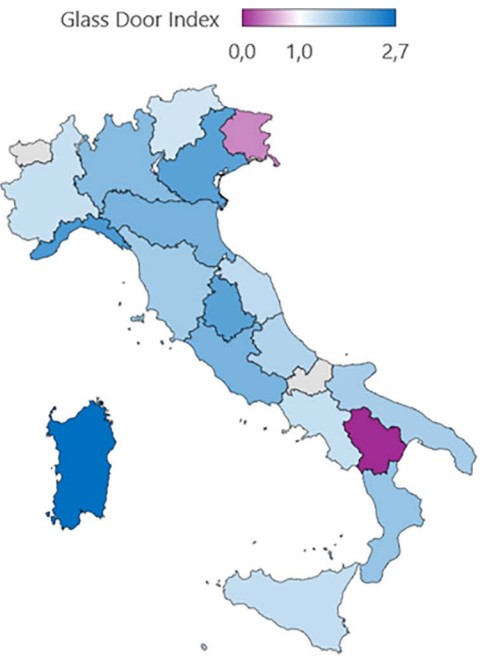

**Fig 4. Gender disparity in Italian Medical Area in 2023 according to Glass Door Index.** Map created using R (version 4.3.2) and the packages rnaturalearth (v0.3.3), rnaturalearthdata (v0.1.0), sf (v1.0-14), and ggplot2 (v3.4.4). Geographic data are from Natural Earth (https://www.naturale-arthdata.com) and are in the public domain. No copyright restrictions apply.

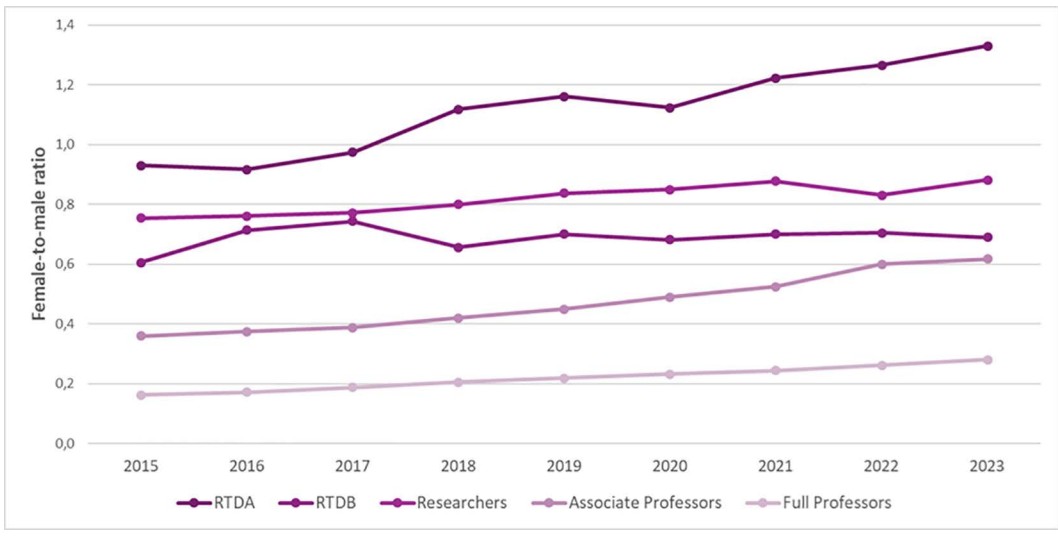

**Fig 5. Temporal trend of female-to-male ratio from 2015 to 2023.**

for these regions – along with Basilicata – the available data did not consistently meet the criteria required for GCI calculation over the full time span considered. Among the remaining regions, median GCI values ranged from 1.6 in Piedmont to 2.9 in Sardinia, while a general tendency toward decreasing vertical segregation emerged over the analysed period. Specifically, Friuli-Venezia Giulia demonstrated a marked improvement, with the GCI decreasing from 4.0 in 2015 to 1.2

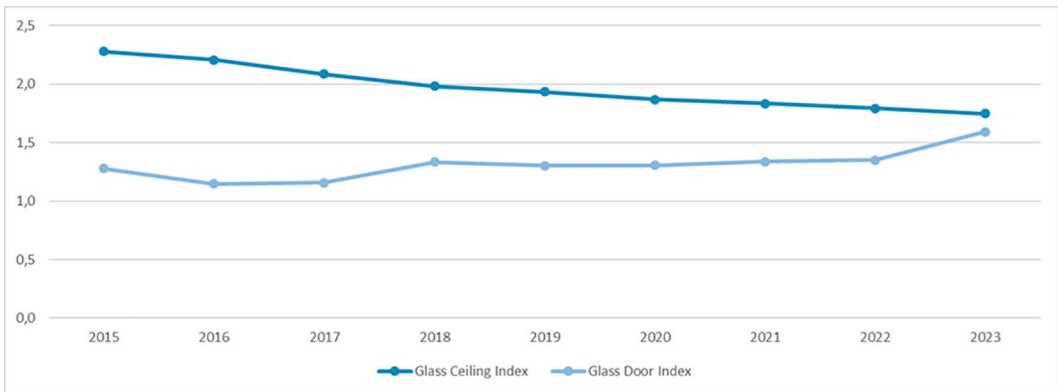

**Fig 6. Temporal trends of gender disparities from 2015 to 2023 according to GCI and GDI.**

in 2023. In contrast, Sardinia represented the only exception, showing a worsening trend in vertical segregation, with the GCI rising from 1.7 in 2015 to 2.9 in 2023.

The GDI values, by contrast, showed heterogeneous trends between 2015 and 2023, with most regions displaying an increased level of disparity disadvantaging women by 2023. Caution is advised when interpreting the indices for regions with a limited number of professionals across academic roles – such as Molise and Trentino-Alto Adige – as small sample sizes may affect the robustness of the estimates. Among the regions with complete data over the entire analysis period, median GDI values ranged from 0.8 in Marche to 2.4 in Friuli-Venezia Giulia. Further details can be found in Figures S1 and S2 in S1 File.

Next, we assessed temporal trends of gender disparities within SSDs according to GDI and GCI. In general, the GCI usually reported a value greater than one, with some exceptions: SSD MED/02 from 2021 to 2023, MED/21 from 2014 to 2016, MED/23 in 2016 and from 2018 to 2020, MED/45 from 2014 to 2023, MED/47 in 2017, and MED/48 in 2014–2015. Among the SSDs with complete data, the highest median GCI value was observed in MED/30 (6.2), while the lowest was recorded in MED/45. For a detailed overview of the trend in each SSD, please refer to Figure S3 in S1 File. When considering the GDI, the situation appeared more heterogeneous, warranting an SSD-by-SSD analysis. Among the SSDs with complete data, the highest median GDI value was found in MED/11 (1.9), while the lowest was observed in MED/35 (0.9). Notably, MED/45 – already characterized by a relatively high proportion of women across academic roles (see paragraph 3.1) – reported a GCI of 0.7 in 2023, consistent with its 2014 value, and a GDI of 2.0, which represents a sharp increase from 0.3 in 2015. Further details are provided in Figure S4 S1 File.

## 4. Discussion

Discrimination against women in the workplace remains a significant challenge, despite ongoing efforts to promote gender equality. According to the European Institute for Gender Equality (EIGE), gender segregation and discrimination persist across various professional fields, with women facing systemic barriers to career advancement and leadership positions [1]. The "Gender Equality Index 2020" report by EIGE highlights that women are underrepresented in decision-making roles and experience a substantial pay gap compared to their male counterparts. This disparity is even more pronounced in sectors such as STEM, where women tend to be concentrated in lower-paying, junior roles, while senior leadership roles remain predominantly male [3]. Additionally, the European Commission's "She Figures 2021" report reveals that women are disproportionately affected by precarious employment conditions and have limited access to research funding and career development opportunities in academia [4]. Scientific evidence further underscores these disparities [19] highlighting the multifaceted nature of these disparities, which manifest in different forms across academic disciplines and

career stages [20–24]. Research also indicates that gender disparities are not uniform across disciplines. However, even in more gender-balanced fields, women tend to face significant barriers when it comes to attaining senior academic positions and receiving recognition for their work [25]. Moreover, research by the European Commission demonstrates that unconscious biases and structural barriers, such as the lack of family-friendly policies and support for work-life balance, contribute to the ongoing discrimination against women in the workplace [6].

The use of the GCI and GDI has allowed us to assess gender disparities in the Italian academic medical field. Specifically, in the context of the Italian academic medical field, these indices serve as valuable tools for identifying barriers that women face in reaching leadership roles and their participation in high-ranking academic positions. The GCI measures the extent to which women face obstacles in advancing to top leadership positions, reflecting the metaphorical "ceiling" that limits their progression. In the context of Italian academia, studies have demonstrated that while women may achieve significant representation at lower academic levels, they tend to be underrepresented in senior roles such as professors, department heads, and deans. The GCI helps quantify this disparity, indicating how the progression of women in academia slows down as they move up the academic hierarchy. On the other hand, the GDI assesses the barriers women encounter when attempting to enter academic fields or institutions, essentially evaluating how open or closed the "door" is to women at the onset of their academic careers. It looks at factors like recruitment practices, access to research funding, and the gendered dynamics of hiring decisions. In the Italian medical academic sector, evidence from the GDI shows that while women are well-represented in medical education, their participation drops in higher positions, which might be influenced by unconscious bias, gendered perceptions of leadership, and institutional structures favouring men. In Italy, these indices have been instrumental in demonstrating that despite various initiatives for gender equality, there remain significant gender gaps in medical academia. The gender imbalances are not only seen in leadership roles but also in areas like research funding allocation and career advancement opportunities. By using the GCI and GDI, these studies highlight the need for policies that promote gender equality at all stages of an academic career—ensuring that women are not only able to enter the field but also have the support and opportunities to thrive and lead in their respective fields. Incorporating these tools into research also opens the door for more targeted interventions. For example, using the GCI and GDI can guide institutions to implement specific gender equality measures, such as mentorship programs, family-friendly policies, and transparent recruitment processes, that aim to dismantle the structural and cultural barriers that contribute to gender disparity in academic medicine. These indices provide a clear and quantifiable way to track progress and design more effective strategies to foster gender equity within the medical academic sector. The aim was to test the hypothesis that, also in this area, the gender gap in academic careers is not going to disappear in a relatively short time. As it has been shown by recent research findings, the issue is not just that of waiting the time needed to see the new generations of women that have entered academia break the glass ceiling. The gender gap is self-sustaining through different mechanisms at different stages of the academic career, as the glass door effect revealed is happening at the transition from temporary positions to the tenure track position, after its recent introduction in the Italian university system [10,18,26,27] by the latest general reform of the Italian university system (Law 240/2010, better known as the Gelmini Reform). Our findings confirm the hypothesis for the medical area, showing that a higher number of women in early career stages does not correspond to a proportionate rate of women in tenure track positions. By contrast, as the percentage of women in the initial stages increases, the presence of obstacles to stabilization becomes more evident. Despite regulatory interventions and gender mainstreaming policies, the career scissor diagram for 2023 clearly illustrates the permanence of the decline in the representation of women as they advance through academic ranks, culminating in a significant underrepresentation at the full professor level. The temporal trend analysis from 2014 to 2023 offers a glimmer of hope, revealing a gradual increase in the female-to-male ratio across academic roles and a reduction in gender disparities in leadership positions. However, the pace of change remains slow, and achieving gender parity is still a distant goal for most academic roles. Furthermore, as Ferree and Purkayastha suggest, the decrease of a disadvantage in the transition from the intermediate (B) to the final level (A), is not unambiguously signalling greater gender equality to the extent that, on the one hand,

women who are at the first level are still less likely to become tenured researchers than their male colleagues and, on the other hand, this decrease could be due precisely to the discrimination that women experience in the previous step [28]. Gender discrimination in the workplace often leads women to delay childbearing, fearing that employers may be less willing to hire, promote, or support working mothers. Many women also choose to leave their jobs after childbirth to care for their children, which further influences their decisions about starting a family. The widespread fear of discrimination against pregnant women and women with young children is closely linked to their choices to postpone or avoid having children [29].

Through the introduction of the GDI, the analysis empirically highlights the existence of gender differentiation processes in academic career paths within the medical area, starting from the recruitment phases. Specifically, the career scissor diagram varies significantly across different disciplines, and the comparison between GDI values before the reform and after the reform shows that, following the Gelmini reform, there has been a systematic relative decrease in the number of women accessing stable academic positions [18]. Comparison between SSDs has revealed disciplines that historically show a predominance of the male gender, especially in surgical specialties. The higher percentage of men in these sectors may be attributed to various historical, cultural, and social factors. Traditionally, many of these fields have been perceived as more technical or less compatible with roles historically assigned to women. This may have led to lower initial participation of women and, consequently, to a persistence of gender disparity. Additionally, structural barriers and gender bias, the lack of female role models, and traditional gender expectations as well as the shortage in services regarding family responsibilities may have contributed to discouraging them from pursuing or continuing careers in these fields. Conversely, other disciplines, such as nursing sciences, show a predominance of women, although horizontal segregation persists.

The observed trend is consistent with broader European patterns, where gender disparities in academia are well-documented. As noted in several European studies, women are often underrepresented in higher-ranking positions, despite their significant contributions in early career stages [4]. For example, in the "She Figures 2021" report, women account for just 29% of full professors in Europe, despite representing nearly half of all doctoral graduates and post-doctoral researchers [4]. The predominance of women in the early career stages, particularly in research-focused roles, may be partly attributed to factors such as greater flexibility in these positions, which are sometimes more accessible for individuals balancing work and family responsibilities. However, as careers progress, the proportion of women drops substantially in response to institutional and cultural barriers. Women's advancement is often slowed by the so-called "leaky pipeline," a phenomenon where women enter academia in equal or greater numbers than men but face increasing barriers that prevent them from reaching senior leadership roles. This includes not only biased career evaluation systems but also the challenges posed by work-life balance issues, insufficient mentorship, and unequal access to professional networks, which are often critical for career advancement [7]. Geographical disparities further compound this issue, with regions like Sardinia, Liguria, and Veneto exhibiting the highest gender inequalities. In contrast, regions such as Piedmont and Trentino-Alto Adige show more equitable distributions, indicating regional variations in the effectiveness of gender equality policies. The GCI and GDI reinforce the notion that systemic barriers remain prevalent, with few regions demonstrating substantial progress in achieving gender parity in recruitment and promotion processes. The temporal trend analysis from 2014 to 2023 offers a glimmer of hope, revealing a gradual increase in the female-to-male ratio across academic roles and a reduction in gender disparities in leadership positions. However, the pace of change remains slow, and achieving gender parity is still a distant goal for most academic roles. As Ferree and Purkayastha suggest, the decrease of a disadvantage between adjacent levels, and more particularly in the transition from the intermediate (B) to the final level (A), does not necessarily mean greater gender equality to the extent that, on the one hand, women who are at the first level are still less likely to become tenured researchers than their male colleagues and, on the other hand, this decrease could be due precisely to the discrimination that women experience in the previous step [28]. The regional distribution of inequalities also suggests possible hypotheses about the gender distribution: women may be overrepresented in the

highest positions in regions that are less desirable for logistic and prestige reasons as the case of Molise (a mountainous region with logistic difficulties, especially during the winter season) may suggest – although small numbers in the data call for cautious interpretations.

The study has also several limitations. First of all, we considered the lack of data on the number of research fellows in the years preceding 2023. As a result, this category has been excluded from the calculation of the GCI and GDI, potentially leading to an incomplete picture of gender inequality in early career stages. Additionally, the lack of data on the different academic positions did not allow for a complete calculation of the GCI and GDI indices, both at the regional level and across the SSDs.

Moreover, the emphasis on quantitative data might overlook qualitative aspects of gender discrimination, such as personal experiences and organizational cultures that also play significant roles in shaping women's career trajectories through context specific social mechanisms [30]. The study highlights that gender disparities are not uniform across disciplines. However, the research could benefit from a more detailed analysis of why certain fields are more gender-balanced than others, which might offer insights into more effective gender equality strategies. While the study mentions unconscious biases and structural barriers and suggests possible contextual mechanisms being at work in different regions, further research could delve deeper into these aspects to offer more concrete recommendations on how to address them [30]. Finally, we acknowledge the importance of a comprehensive and current definition of "gender". According to the World Health Organization (WHO), "gender refers to the characteristics of women, men, girls and boys that are socially constructed" [31]. This encompasses the norms, behaviours, roles, and relationships associated with being a woman, man, girl, or boy, as well as the opportunities and attributes that society considers appropriate for each [32]. However, in this study, the classification of individuals was based on the Cineca portal's data, which records only legal sex (male and female). Therefore, the term "gender" in this manuscript reflects a binary classification and does not account for self-identified or non-binary gender identities.

## 5. Conclusion

In conclusion, while there have been notable advancements in gender equality in the Italian academic medical field, persistent inequalities continue to hinder the full integration and progression of women in the sector. These inequalities are evident not only in the underrepresentation of women in leadership roles, such as senior professors, heads of departments, and academic deans, but also in the unequal distribution of research funding and access to career advancement opportunities. Despite various initiatives aimed at promoting gender equality, these disparities remain entrenched due to both visible and invisible structural barriers that continue to favour male counterparts. Our findings suggest that while the glass ceiling remains a significant issue, there is also a pressing need to address the *glass door* phenomenon—the barriers preventing women from entering academic medicine in the first place. Factors such as unconscious bias in recruitment practices, the prioritization of male-dominated networks, and societal expectations surrounding gender roles contribute to this uneven playing field. These issues require a multi-faceted response that targets both the entry-level barriers and the challenges faced by women as they progress through their academic careers. More robust and targeted strategies are necessary to dismantle these structural barriers. Institutional policies should focus not only on ensuring fair and equitable recruitment practices but also on promoting transparent and inclusive promotion processes. This involves revising performance evaluation metrics that traditionally favour male-dominated research areas and ensuring that caregiving responsibilities, which disproportionately fall on women, are not seen as obstacles to career progression. Mentorship programs, networking opportunities, and leadership training tailored specifically for women can also serve as crucial steps in addressing the systemic biases that persist in the field. Furthermore, fostering an inclusive academic environment requires cultural shifts within academic institutions. This includes promoting work-life balance policies, supporting parental leave, and creating family-friendly working conditions to reduce the gendered expectations placed on women. These cultural changes should be coupled with comprehensive policy implementation and sustained monitoring to track progress

toward achieving gender equality. The road to achieving lasting gender equality in Italian academic medicine is undeniably challenging, as it requires overcoming deeply rooted biases, norms, and practices. However, it is a crucial goal that will not only benefit women but also enrich the academic medical community as a whole. Studies consistently show that diversity in leadership fosters better decision-making, more innovative research, and improved outcomes across various sectors, including healthcare. Therefore, addressing gender inequality is not just a matter of social justice but also an imperative for improving the quality of education and research in the medical field. Achieving true gender equality requires continuous, multi-dimensional efforts that include policy reforms, cultural transformation, and institutional support. In this context, political actors also have a key role to play by promoting and implementing equity policies that support these objectives at national and regional levels. The success of these efforts will depend on the willingness of institutions to recognize and address the systemic biases that perpetuate inequality, ultimately creating a more inclusive and equitable environment that allows all individuals, regardless of gender, to reach their full potential in academia.

## Supporting information

**S1 File. Supporting figures and tables.**
(DOCX)

## Acknowledgments

The authors thank editor and our anonymous reviewers for their important feedback and guidance. The authors also thank the Gender Prevention Working Group of the Italian Society of Hygiene, Preventive Medicine, and Public Health – SItI.

## Author contributions

**Conceptualization:** Roberta Magnano San Lio, Andrea Maugeri, Maria Carmela Agodi, Ilenia Picardi, Antonella Agodi.

**Data curation:** Roberta Magnano San Lio, Andrea Maugeri, Ilenia Picardi.

**Formal analysis:** Roberta Magnano San Lio, Mara Morini, Enrico Di Rosa, Alessandra Sinopoli, Virginia Casigliani, Manuela Martella, Andrea Maugeri, Martina Barchitta, Maria Carmela Agodi, Ilenia Picardi, Antonella Agodi.

**Supervision:** Antonella Agodi.

**Writing – original draft:** Roberta Magnano San Lio, Andrea Maugeri.

**Writing – review & editing:** Roberta Magnano San Lio, Mara Morini, Enrico Di Rosa, Alessandra Sinopoli, Virginia Casigliani, Manuela Martella, Andrea Maugeri, Martina Barchitta, Maria Carmela Agodi, Ilenia Picardi, Antonella Agodi.

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
