## [Decision Letter · Decision Letter 0]

Dear Dr. Agodi,

Thank you for submitting your manuscript to PLOS ONE. After careful consideration, we feel that it has merit but does not fully meet PLOS ONE’s publication criteria as it currently stands. Therefore, we invite you to submit a revised version of the manuscript that addresses the points raised during the review process.

We look forward to receiving your revised manuscript.

Kind regards,

Claudia Noemi González Brambila, Ph.D.

Academic Editor

PLOS ONE

2. In the online submission form, you indicated that [The data underlying the results presented in the study are available from the corresponding author].

Additional Editor Comments (if provided):

Reviewers' comments:

Reviewer's Responses to Questions

**Comments to the Author**

1. Is the manuscript technically sound, and do the data support the conclusions?

Reviewer #1: Yes

Reviewer #2: Partly

Reviewer #3: Yes

Reviewer #4: Yes

2. Has the statistical analysis been performed appropriately and rigorously?

Reviewer #1: Yes

Reviewer #2: N/A

Reviewer #3: N/A

Reviewer #4: Yes

3. Have the authors made all data underlying the findings in their manuscript fully available?

Reviewer #1: Yes

Reviewer #2: Yes

Reviewer #3: Yes

Reviewer #4: Yes

4. Is the manuscript presented in an intelligible fashion and written in standard English?

Reviewer #1: Yes

Reviewer #2: Yes

Reviewer #3: Yes

Reviewer #4: Yes

Reviewer #1: Although the study is well conducted and the manuscript written in intelligible way, there are parts that would make it better for the reader to understand methods and results better. I have added those explicitly in the attached file. Thanks.

Reviewer #2: This is potentially an interesting manuscript in that there is evidence that fewer women are making it into the higher academic positions related to medicine and all of its related subspecialties across Italy. There is also some evidence that there is also some regional variation in these gender ratios as well.

A number of significant issues have been identified with the manuscript in its current form which require the further attention of the authors. This includes-

1) An absence of numerical data elements in the results section of the abstract

2) An extremely long Introduction section which needs to be condensed down with some of the relevant information being used for the Discussion section (to prevent repeating the same information in the Discussion section, which is apparent in places).

3) There is a failure to provide the summary data for the actual numbers of females versus males for the various medical subspecialties with instead only the data for the ratios being provided. It would be useful to provide this summary data along with the gender ratios so that the reader could more readily ascertain as to where real progress is being made (ie within either the subspecialties or the regions with higher numbers of medical professionals overall). This would be pertinent to Figure 1, Figure 2 and Figure 5. As the authors allude to on lines 288-289 low numbers of medical professionals in some regions mean that the results of the GDI values have to be treated with caution. Plus, there is also the issue with missing data elements for the GDI and the GCI (see below).

4) A number of the Tables/Figures are too crowded/contain too much data to appear in the main manuscript. For eg Table 1, Figures 7-10.

5) Plus of note there seem to be missing data elements for both the GCI and the GDI in Figures 7-10. Are these missing data elements excluded from any further consideration in the results? Plus does this not limit the validity of the GCI/GDI calculations? Hence this raises the question as to how reliable the summary data that is contained in the graphics in Figure 3 and Figure 4 actually is. It seems that the only data which can be reliably depicted/summarized for either the CGI or GDI is where there is a complete set for either the GCI or GDI per year either for the subspecialty or for the region of interest. It is this subset of the data that needs to be focused on in the results (with a median and range being reported of either the overall GCI or GDI score per year for all of the study group and then for each region-if there is a complete set of data)

Reviewer #3: The aim of this manuscript is to examine gender inequalities in top positions and career progression within Italy's Medical Sciences sector (Area 06) from 2014 to 2023, through data from the Cineca portal, including details on academic roles, gender, region, and scientific-disciplinary (SSDs).

This manuscript is well-structured, informative, and supported by relevant data and sources. However, there are a few suggestions for improvement regarding clarity, and conciseness. For these reasons, the manuscript requires major changes.

Please find below an enumerated list of comments on my review of the manuscript:

MINOR POINTS:

The authors should provide a list of the abbreviations, mentioned in this manuscript.

MAJOR POINTS:

INTRODUCTION:

LINE 50: The term “gender” indicates the identities of male, female, and gender-diverse populations in a social context. The World Health Organization’s (WHO) definition of gender refers to “the roles, behaviors, activities, attributes, and opportunities that any society considers appropriate for girls and boys, and women and men”(see, for reference: https://doi.org/10.3390/healthcare11060828). This is the major concern of this manuscript: this manuscript will benefit from providing an organic definition of the term “gender”, in according to WHO and recent scientific evidence on this topic.

LINE 60: Furthermore, there is a strong connection between gender differences and STEM (Science, Technology, Engineering, and Mathematics), which manifests in multiple ways, including representation, career advancement, biases, and societal perceptions (see, for reference: https://doi.org/10.1080/2331186X.2024.2439655).

As regards the originality and strengths of this manuscript, this is a significant contribute to the ongoing research on this topic, as it provides a broad overview of gender disparities in academia, specifically within the European and Italian contexts. Overall, the contents are rich, and the authors also give their deep insight for some works.

Moreover, the discussion of this manuscript effectively connects gender disparities to structural and cultural barriers.

The conclusion of this manuscript is perfectly in line with the main purpose of the paper: the authors have designed and conducted the study properly. As regards the conclusions, they are well written and present an adequate balance between the description of previous findings and the results presented by the authors.

Finally, this manuscript also shows a basic structure, properly divided and looks like very informative on this topic. Furthermore, figures and tables are complete, organized in an organic manner and easy to read.

In conclusion, this manuscript is densely presented and well organized, based on well-synthetized evidence. This manuscript effectively outlines the persistent issue of gender inequality in academia, within Europe and Italy. It is well-researched, citing relevant data and discussing structural barriers such as financial disparities, and limited mentorship opportunities. The authors were lucid in their style of writing, making it easy to read and understand the message, portrayed in the manuscript. Besides, the methodology design was appropriately implemented within the study. However, many of the topics are very concisely covered. This manuscript provided a comprehensive analysis of current knowledge in this field. Moreover, this research has futuristic importance and could be potential for future research. However, major concerns of this

manuscript are with the introductive section: for these reasons, I have major comments for this section, for improvement before acceptance for publication. The article is accurate and provides relevant information on the topic and I have some major points to make, that may help to improve the quality of the current manuscript and maximize its scientific impact. I would accept this manuscript if the comments are addressed properly.

Reviewer #4: The manuscript is scientifically sound. There is no mention of getting any IRB approval. It seems like its not needed or applicable here. Kindly check with the authors regarding getting any IRB approval.

---

## [Author Response · Author response to Decision Letter 1]

14 Apr 2025

Dear Editor,

We would like to thank you and the reviewers for the time and effort dedicated to evaluating our manuscript titled " Gender Differences in the Italian Academic Landscape: Examining Inequalities within the Medical Area in the Last Decade". We greatly appreciate the insightful and constructive comments, which have helped us improve the clarity, structure, and scientific value of our work.

Below, we provide a point-by-point response to each reviewer’s comments. All changes have been incorporated in the revised version of the manuscript, which we resubmit for your consideration.

Reviewer #1

Comment (C): Although the study is well conducted, and the manuscript written in an intelligible way, there are parts that would make it better for the reader to understand methods and results better. I have added those explicitly in the attached file. Thanks.

Answer (A): We are grateful to Reviewer 1 for this comment and for the suggested improvements. We have revised the manuscript accordingly. Some parts of the introduction have been removed in line with feedback from other reviewers. The “Integrazione del decreto del Ministro dell’istruzione, dell’università e della ricerca 1 settembre 2016, n. 662” could not be translated into English, as it refers to a specific Italian legal document. In the Results section, we chose not to include percentages for all specialties with male predominance, to maintain clarity and readability. However, we have made the table more reader-friendly, as suggested. In the Discussion, we added a comment and reference on the challenges women face in balancing family and academic careers, particularly regarding decisions around motherhood. We also removed redundant sentences to avoid overlap with the Introduction. Lastly, references were not included in the Conclusions section, in line with standard academic conventions.

Reviewer #2

C: This is potentially an interesting manuscript in that there is evidence that fewer women are making it into the higher academic positions related to medicine and all of its related subspecialties across Italy. There is also some evidence that there is also some regional variation in these gender ratios as well.

A: We thank Reviewer 2 for this overall positive assessment.

C: There is an absence of numerical data elements in the results section of the abstract.

A: We apologize for the lack of clarity in the abstract. Numerical elements have now been included in the revised abstract to enhance its informative value.

C: An extremely long Introduction section which needs to be condensed down with some of the relevant information being used for the Discussion section (to prevent repeating the same information in the Discussion section, which is apparent in places).

A: We fully agree and have significantly revised and shortened the Introduction. Relevant content has been redistributed or removed to reduce repetition with the Discussion section.

C: There is a failure to provide the summary data for the actual numbers of females versus males for the various medical subspecialties with instead only the data for the ratios being provided. It would be useful to provide this summary data along with the gender ratios so that the reader could more readily ascertain as to where real progress is being made (ie within either the subspecialties or the regions with higher numbers of medical professionals overall). This would be pertinent to Figure 1, Figure 2 and Figure 5. As the authors allude to on lines 288-289 low numbers of medical professionals in some regions mean that the results of the GDI values have to be treated with caution. Plus, there is also the issue with missing data elements for the GDI and the GCI (see below).

A: We apologize if this part of our analysis was unclear. As requested, we have now included absolute numbers for each academic position for the year 2023 at the beginning of the Results section. Additionally, in the Supplementary Information, we have provided two tables stratified by SSD and by region. Data for other years remain available through the Cineca portal (https://cercauniversita.mur.gov.it).

However, we believe that including absolute numbers for the time trend graphs would reduce readability without adding substantial value. The use of proportions, ratios, and calculated indices provides a clearer and more standardized interpretation of gender disparities, which might otherwise be obscured by raw numbers.

C: A number of the Tables/Figures are too crowded/contain too much data to appear in the main manuscript. For eg Table 1, Figures 7-10.

A: We appreciate this suggestion. To improve the manuscript’s readability, we have moved Table 1 and Figures 7–10 to the Supplementary Information.

C: Plus of note there seem to be missing data elements for both the GCI and the GDI in Figures 7-10. Are these missing data elements excluded from any further consideration in the results? Plus does this not limit the validity of the GCI/GDI calculations? Hence this raises the question as to how reliable the summary data that is contained in the graphics in Figure 3 and Figure 4 actually is. It seems that the only data which can be reliably depicted/summarized for either the CGI or GDI is where there is a complete set for either the GCI or GDI per year either for the subspecialty or for the region of interest. It is this subset of the data that needs to be focused on in the results (with a median and range being reported of either the overall GCI or GDI score per year for all of the study group and then for each region-if there is a complete set of data)

A: Thank you for highlighting this important point. We would like to clarify that the elements in question are not technically "missing." Rather, in some instances, the data available did not meet the conditions necessary for calculating the indices. As clarified in the revised Methods and Results sections, the GCI and GDI could not be computed in regions or SSDs where key academic roles were absent (e.g., no full professors or no RTDA/RTDB positions).

We now specify in the manuscript that summary statistics (e.g., medians and ranges) are presented only for subsets with complete data. These limitations have also been explicitly acknowledged in the revised Discussion section.

Reviewer #3

C: The aim of this manuscript is to examine gender inequalities in top positions and career progression within Italy's Medical Sciences sector (Area 06) from 2014 to 2023, through data from the Cineca portal, including details on academic roles, gender, region, and scientific-disciplinary (SSDs). This manuscript is well-structured, informative, and supported by relevant data and sources. However, there are a few suggestions for improvement regarding clarity, and conciseness. For these reasons, the manuscript requires major changes. Please find below an enumerated list of comments on my review of the manuscript.

A: We thank Reviewer 3 for this kind and constructive evaluation.

C: The authors should provide a list of abbreviations.

A: We have now included a list of abbreviations at the end of the manuscript, before the References section.

C: The term “gender” indicates the identities of male, female, and gender-diverse populations in a social context. The World Health Organization’s (WHO) definition of gender refers to “the roles, behaviors, activities, attributes, and opportunities that any society considers appropriate for girls and boys, and women and men”(see, for reference: https://doi.org/10.3390/healthcare11060828). This is the major concern of this manuscript: this manuscript will benefit from providing an organic definition of the term “gender”, in according to WHO and recent scientific evidence on this topic.

A: We thank the reviewer for raising this point. However, we are unable to adopt the broader WHO definition of “gender” as our data source (the Cineca portal) categorizes individuals strictly by legal sex: male (M) and female (F). As such, our use of the term “gender” reflects this binary classification and does not include self-identified gender identities.

C:, there is a strong connection between gender differences and STEM (Science, Technology, Engineering, and Mathematics), which manifests in multiple ways, including representation, career advancement, biases, and societal perceptions (see, for reference: https://doi.org/10.1080/2331186X.2024.2439655).

A: We appreciate this suggestion and have revised the Introduction accordingly to include this important aspect, along with the suggested reference.

C: As regards the originality and strengths of this manuscript, this is a significant contribute to the ongoing research on this topic, as it provides a broad overview of gender disparities in academia, specifically within the European and Italian contexts. Overall, the contents are rich, and the authors also give their deep insight for some works. Moreover, the discussion of this manuscript effectively connects gender disparities to structural and cultural barriers.

A: We sincerely thank Reviewer 3 for the encouraging feedback and support for our study.

C: However, major concerns of this manuscript are with the introductive section: for these reasons, I have major comments for this section, for improvement before acceptance for publication. The article is accurate and provides relevant information on the topic and I have some major points to make, that may help to improve the quality of the current manuscript and maximize its scientific impact. I would accept this manuscript if the comments are addressed properly.

A: We have thoroughly revised the Introduction based on all reviewers’ feedback, improving its clarity, focus, and conciseness.

Reviewer #4

C: The manuscript is scientifically sound. There is no mention of getting any IRB approval. It seems like its not needed or applicable here. Kindly check with the authors regarding getting any IRB approval.

A: We confirm that ethical approval was not required for this study, as all data were obtained from the publicly accessible Cineca portal. The data are anonymized and presented in aggregated form, in compliance with ethical research standards.

Once again, we sincerely thank all reviewers for their valuable contributions. We hope the revised version meets the journal’s standards and look forward to your feedback.

Kind regards,

Antonella Agodi

---

## [Decision Letter · Decision Letter 1]

https://plos.org/protocols?utm_medium=editorial-email&utm_source=authorletters&utm_campaign=protocols

2. Is the manuscript technically sound, and do the data support the conclusions?

3. Has the statistical analysis been performed appropriately and rigorously?

4. Have the authors made all data underlying the findings in their manuscript fully available?

5. Is the manuscript presented in an intelligible fashion and written in standard English?

what does this mean?

**Do you want your identity to be public for this peer review??>**

---

## [Author Response · Author response to Decision Letter 2]

14 May 2025

Dear Editor,

We would like to thank you and the reviewers for the time and effort dedicated to evaluating our manuscript titled " Gender Differences in the Italian Academic Landscape: Examining Inequalities within the Medical Area in the Last Decade". We greatly appreciate the insightful and constructive comments, which have helped us improve the clarity and scientific value of our work.

Below, we provide a point-by-point response to each reviewer’s comments. All changes have been incorporated in the revised version of the manuscript (in blue), which we resubmit for your consideration.

Reviewer #1

Comment (C): The manuscript has been improved substantially with full description of key topics and interpretations. I have only found a few areas where minor corrections may be required to make the article easy for the reader.

Answer (A): We are grateful to Reviewer 1 for this comment and for the suggested improvements. We have revised the manuscript accordingly.

Reviewer #3

C: Dear Authors, as much as I can understand Your position, I strongly advise You to use the gender definition adopted by WHO. even though You cannot adapt to the italian system You should underline this. please use all the provided references also to improve your discussion.

A: We thank Reviewer 3 for this comment. As recommended, we have included this point and the suggested references in the discussion section.

Once again, we sincerely thank all reviewers for their valuable contributions. We hope the revised version meets the journal’s standards and look forward to your feedback.

Kind regards,

Antonella Agodi

---

## [Editor Report · Decision Letter 2]

Gender Differences in the Italian Academic Landscape: Examining Inequalities within the Medical Area in the Last Decade

PONE-D-25-05019R2

Dear Dr. Agodi,

We’re pleased to inform you that your manuscript has been judged scientifically suitable for publication and will be formally accepted for publication once it meets all outstanding technical requirements.

Kind regards,

Claudia Noemi González Brambila, Ph.D.

Academic Editor

PLOS ONE
---

## [Editor Report · Acceptance letter]

PONE-D-25-05019R2

PLOS ONE

Dear Dr. Agodi,

I'm pleased to inform you that your manuscript has been deemed suitable for publication in PLOS ONE. Congratulations! Your manuscript is now being handed over to our production team.

Kind regards,

on behalf of

Dr. Claudia Noemi González Brambila

Academic Editor

PLOS ONE